# GDReBase: A Knowledge Base for Relations between Human Gut Microbes and Diseases Based on Deep Learning

Haolei Xu ⓘ, Xin Li, Xiaolong Dai ⓘ, Chunhao Liu, Dongxiao Wang, Chenghao Zheng, Kaihua Liu ⓘ, Sitong Liu, Yufei Zeng, Ziyang Song, Shanzhu Cui ⓘ and Yongdong Xu *ⓘ

School of Computer Science and Technology, Harbin Institute of Technology at Weihai, Weihai 264209, China
* Correspondence: ydxu@hit.edu.cn; Tel.: +86-186-6319-9194

**Abstract:** Gut microbes play a prominent role in many aspects of human health, as seen through the increasing number of related studies. The accumulation of intestinal-flora-related studies enables us to better understand the various relationships between human gut microbes and other factors that affect the human body. However, the existing database does not meet the requirements of scientists to browse or retrieve the latest and most comprehensive published data. Thus, a knowledge base containing data related to gut microbes with updates occurring in real time would be highly valuable. We present a knowledge base of consistently curated relationships between human gut microbes and disease. By continuously and automatically collecting papers published in mainstream journals and using deep learning and NLP methods for entity relationship identification, GDReBase has now integrated 3674 diseases, 687 microbes, 7068 relationships, and 13,553 pieces of evidence from 518,286 papers, a figure that will continue to grow. GDReBase is a convenient and comprehensive resource for gut microbiology research and can be accessed free of charge.

**Keywords:** gut microbes; diseases; knowledge database; deep learning; NLP

## 1. Introduction

The human gut microecosystem contains trillions of microbes that constitute the largest and most complex ecological community in the human body, thus playing a prominent role in digesting food, extracting nutrients, and defending against intestinal pathogens [1]. Gut microbes and their metabolites are inextricably linked to the regulation of human physiological homeostasis and the occurrence of disease, which are essential for human health [2]. Many studies have shown that gut microbes are closely associated with a variety of physical and mental health problems. For example, research shows that gut microbes are involved in obesity and metabolic disorders [3]. Obesity is associated with the relative abundance of dynamics of several bacterial divisions [4–6]. In addition, other studies have reported that gut microbes contribute to a number of immune inflammatory conditions, including inflammatory bowel disease [7,8], rheumatoid arthritis [9], and multiple sclerosis [10]. Moreover, some studies have found that gut microbial dysfunction may have an association with a number of neurodegenerative diseases, such as Alzheimer's disease [11], Parkinson's disease [12], and amyotrophic lateral sclerosis [13,14]. Further studies have also shown that tumorigenesis [15], the effectiveness of immunotherapy [16,17], and prognosis [18] are also associated with gut microbes.

Various human gut metagenome profiles are accumulating due to the rapid development of high-throughput metagenomic sequencing technologies. These profiles provide useful assistance in the biological processes regulated by gut microbes. In some pioneering studies, resources of raw sequencing data have been constructed for storage. Examples include the National Center for Biotechnology Information (NCBI) Sequence Read Archive (SRA) [19], the European Nucleotide Archive (ENA) [20], and the DNA Data Bank of Japan (DDBJ) [21]. Meanwhile, several databases on gut microbes have been developed in

previous studies. Belman et al. [22] developed MuPhenome, Janssens et al. [23] introduced Disbiome, Oliveira et al. [24] created MicrobiomeDB, and Wu et al. [25] curated the Gut Microbes Database data repository (GMrepo). These existing resources provide GDReBase with a dictionary of gut microbial names, greatly improving the reliability of GDReBase data. Due to the rapid development of bioinformatics, a large amount of knowledge appears in numerous papers every day, making manual access to such vast amounts of knowledge in real time impractical. Although various databases are capable of storing data on human gut microbes and providing a variety of functions for researchers to access and analyze the target data, we note that most of the existing databases are manual. Their data volumes and coverage are often inadequate and not up to date. Our goal is to obtain a constant flow of information about new and possible relationships between gut microbes and human disease from mainstream and scientific journals. Users can use GDReBase to view the latest research findings.

To provide fast and intuitive access to steadily growing intestinal microbial information, we introduce a comprehensive, indexed, and updated knowledge base (GDReBase) for relations between human gut microbes and diseases. In total, data on the relationship between gut microbes and disease are integrated into GDReBase. The information in GDReBase has been carefully organized for ease of use and implemented with browsing and search functions. In contrast to the existing databases related to gut microbes, GDReBase automatically extracts accurate data from relevant papers that are selected. Researchers can access the most up-to-date data from GDReBase and compare their results with published studies. GDReBase is a convenient and comprehensive resource for gut microbiology research and can be accessed free of charge at http://www.gdrebase.com.

## 2. Results

### 2.1. Database Content and Statistics

As a database dedicated to gut microbe research, GDReBase automatically extracts the relationship between gut microbes and disease from screened papers and updates them every month to provide more comprehensive microbe disease knowledge for gut microbe researchers.

Up to August 2022, we searched through a total of 518,286 papers in the 368 journals selected. GDReBase currently obtains 7068 relationships and their corresponding 13,553 pieces of evidence. Figure 1 shows the variation in the number of papers crawled and the evidence extracted by year. As can be seen, the number of papers essentially followed a steady growth trend, despite a decline in 2020, probably due to a shift in researcher interest due to COVID-19. The top eight diseases and microbes that appeared most frequently in the evidence are listed in Figure 2a,b. Our disease entity identification model obtained an F1 score of 90.12% for NCBI diseases [26], an improvement of 3.75 percentage points compared to the baseline model BERT BASE [27]. In practical applications, we randomly sampled 200 pieces of evidence and found that 12 of them contained misidentified disease entities and another four were unable to confirm a relationship between microbes and diseases. Fortunately, most of the errors had a pattern. For example, the model identified some words ending in 'virus' as diseases, and some microbes were also treated as diseases. These errors can be eliminated by rules and will not affect the normal use of GDReBase.

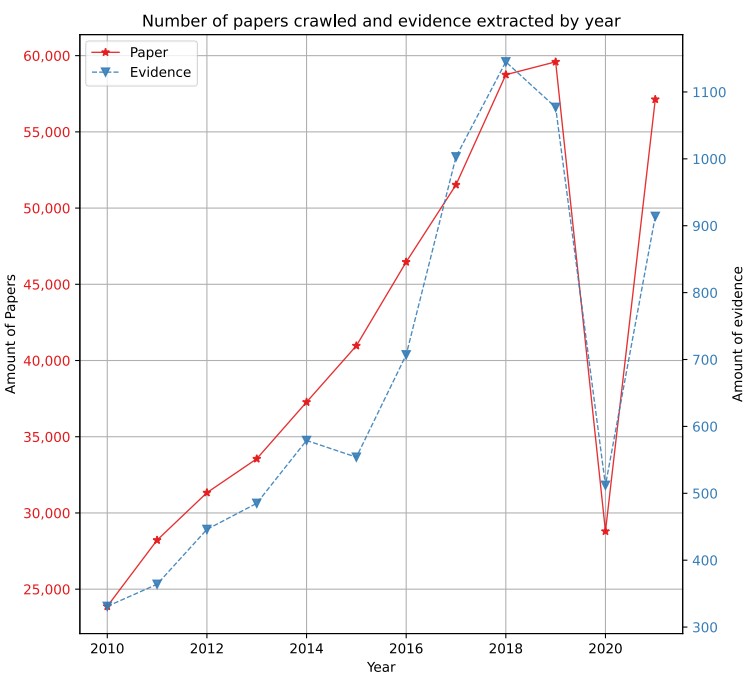

**Figure 1.** Number of papers crawled, and evidence extracted by year.

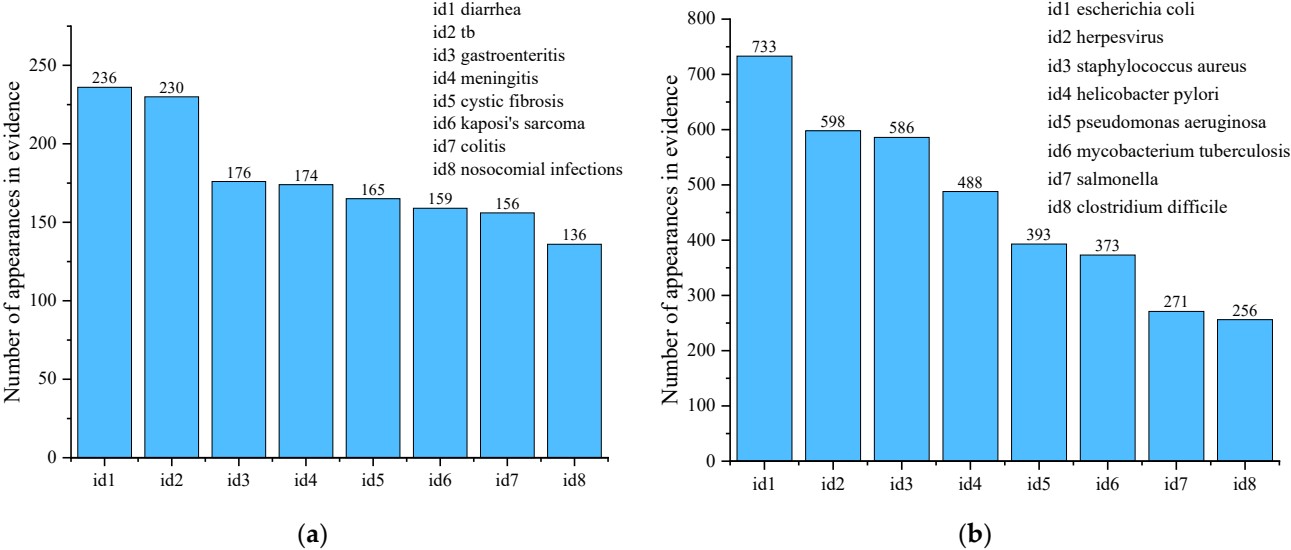

**Figure 2.** (**a**) Top 8 diseases with most evidence; (**b**) top 8 microbes with most evidence.

*2.2. Web Interface*

GDReBase provides a search engine for querying detailed information on gut microbes and diseases from our integrated resources. As shown in Figure 3, in the 'Search View', users can click on 'Entity Search' to search for a particular disease or microbe, or click on 'Relation Search' to search for a relationship between a pair of entities. When the user types, a fuzzy search prompt is given at the bottom of the input box. It should be noted that all letters in GDReBase are in lowercase form, and users must obey this when using it. After submitting an entity, the system can retrieve and return results related to the entity. Users can click on the hyperlinks in the search results to see 'Table view' and 'Graph view'. In 'Table view', users can clearly see connections between searched entities and corresponding evidence. If users want further information, they can click on the hyperlink of 'ReferenceID' to view details of the paper. We also provide the URL of paper where the evidence originated for readers in 'Detail view'. Since a relationship may have multiple

pieces of evidence, we de-duplicated the data in the 'Table view' and provided a visual representation of the entities with relationships in the 'Graph view', which is essentially an undirected graph. Users can click on a node to zoom in and out of the graph. On the top-right corner of the home page, users can click on the download button to access all the evidence and papers we have collected.

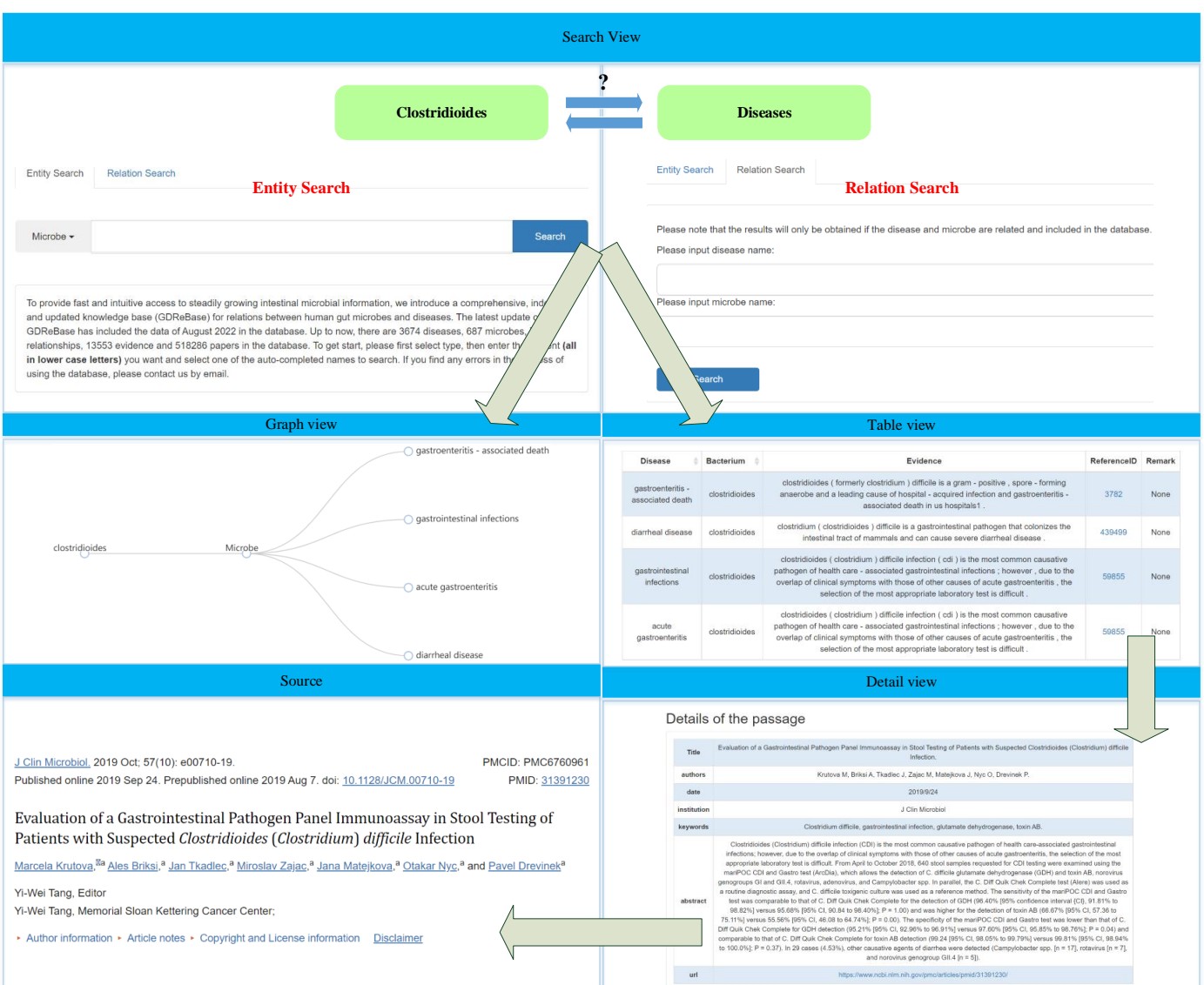

**Figure 3.** Different views of web pages and GDReBase usage.

### 2.3. Automatic Updates

We deployed an automatic update script on the server. The script was started automatically at regular intervals. The script first automatically crawled new papers published in given journals in a recent period and compared results with the table of papers in GDReBase. Results with identical titles and authors were not newly included in GDReBase. After crawling, the script automatically inputted paper texts into NER and RE models to mine the entity relationships. The results were de-duplicated and incorporated into GDReBase.

### 3. Discussion

In terms of research in recent years, databases for gut microbes and diseases have their own focus. Janssens et al. [23] proposed Disbiome as a presentation of published

information on a microbiome diseases database. MicrobiomeDB [24], created by Oliveira et al., was committed to providing a data discovery and analysis platform that allowed researchers to make the most of experimental variables to interrogate microbiome datasets. In addition, Wu et al. [25] developed the Gut Microbes Database (GMrepo), which was dedicated to linking phenotypes to gut microbial composition. These advances provided an outstanding resource for microbiome research. However, the problem is that these databases can be relatively homogenous in terms of data sources and relatively narrow in coverage, and updating is labor intensive.

Compared with the mentioned databases, our database has a strong update capability and efficiency, which can automate the whole process from searching to entity relationship mining. With thousands of new papers published every year, it would be impractical to read them one by one, let alone to learn more about the relationship between gut microbes and diseases. Therefore, GDReBase can help researchers read papers or compare their own results with the latest papers more efficiently, improving work efficiency. Table 1 shows the comparison of various indicators of different databases.

**Table 1.** Comparison with other databases.

| Database | Update | Type | Update Speed | Cover Area |
|---|---|---|---|---|
| Disbiome | √ | Manual | >One month | Published information |
| MicrobiomeDB | √ | Manual | Three months | Microbiome datasets |
| GMrepo | × | Manual | × | Human gut metagenome projects |
| GDReBase | √ | Automatic | Dynamic | Published information |

Inevitably, there may be some limitations to our study. When filtering entity relationships, we focus on accuracy rather than recall to ensure the reliability of GDReBase. There is still room for improvement in the accuracy of using deep learning methods to identify entities and their relationships. As shown in Table 2, words with interrogative or negative meanings in sentences can affect the filtering of entity relationships. In addition, a few entities that are not obviously related still slip through the net. To minimize the trouble caused to the researcher by incorrect relationships, the context of entity relationships was provided to help confirm reliability. At the same time, while limited to the method of paper acquisition, most of the entity relationships we excavated came from the title and abstract of the paper. The number of relationships mined would be greatly increased if the main body of the paper was accessible.

**Table 2.** Some examples of problematic results.

| Example | Problem |
|---|---|
| The role of enteropathogenic <m>escherichia coli</m> epec as a cause of <d>diarrhoea</d> in cancer and immunocompromised patients is ***controversial***. | controversial |
| An increase in <m>aeromonas</m> ***may*** be closely related to the development of <d>enteritis</d>. | may |
| A <m>bacillus</m> calmette guerin bcg model was also established to assess the diagnosis of <d>tuberculosis infection</d> using ec skin test. | no apparent association |

To serve as a useful and convenient data resource for gut microbes, there is still much work to be done for improvement. First, the reliability of the data could be further improved. We intend to use a graph convolutional neural (GCN) network to replace the existing RE model. This approach can achieve better results after initial testing. Furthermore, we intend to use knowledge graph reasoning to further mine potential new relationships between gut microbes and diseases on the basis of the GDReBase, which may provide scientists with some new directions for research. Before building a knowledge graph for reasoning, we will label a large dataset to segment the relationships between diseases and microbes. Overall,

although improvements remain to be achieved, GDReBase can serve as a comprehensive resource to provide intuitive data evidence for gut-microbe-related research.

## 4. Methods

The overall flow chart of constructing the GDReBase is shown in Figure 4.

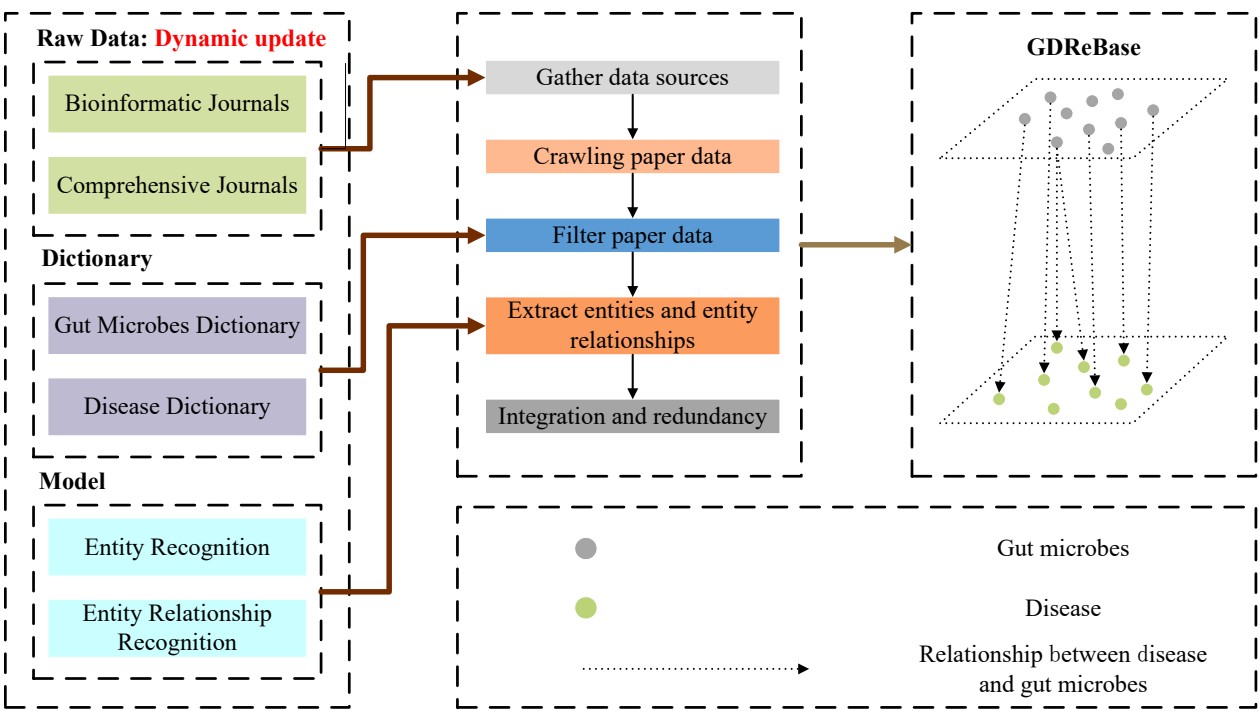

**Figure 4.** The schematic flow of the GDReBase construction.

### 4.1. Overview of Innovativeness

Our method can continuously and automatically collect papers published in main-stream journals and mine relationships between human gut microbes and disease by using deep learning and NLP methods. In the model, we integrated word embedding, character-based representation, and dictionary-matching representation. In addition, we introduced an advanced attention mechanism to further improve the recognition accuracy.

### 4.2. Data Sources and Crawling

On the basis of the Annual Report for World Journal Impact Index (https://wjci.cnki. net/home/Index) (accessed on 27 November 2022), we collected bioinformatic, medical, microbiological, and comprehensive journals with high-impact factors. The detailed list of journals can be downloaded from the GDReBase website. We crawled through the Web of Science and NCBI for titles, abstracts, and accessible texts.

### 4.3. Data Filtering and Relationship Dataset

To filter out papers closely related to gut microbes and disease, we download the classification of gut microbes in NCBI and searched through Malacards (https://www. malacards.org/categories) (accessed on 27 November 2022) for information related to disease. These results are used as a dictionary to match titles and abstracts, as well as papers that meet the requirements (microbes $\geq$ 1, diseases $\geq$ 1) that are screened as sources of papers for entity relationships in GDReBase.

Since it is currently difficult to find datasets or benchmarks for disease microbial relationships, we carried out manual annotation. To speed up the annotation, we first focused on marking the corpus containing keywords, such as 'cause' and 'trigger', and then checked them manually.

### 4.4. NER and RE

To obtain the relationships between the diseases and gut microbes mentioned in the papers, we adopted a deep learning approach for NER and RE. The architecture of the model is shown in Figure 5.

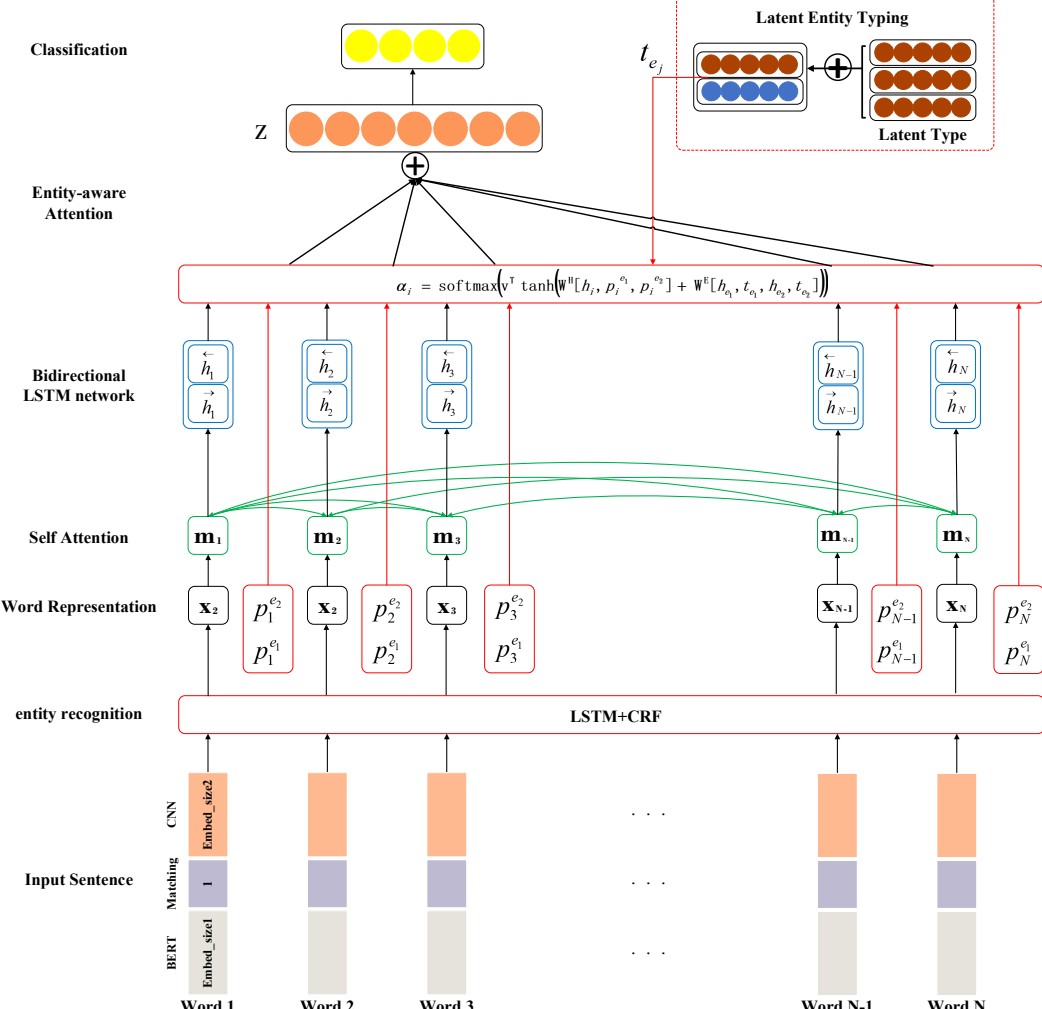

**Figure 5.** The architecture of our model.

After reading a large number of papers, we found that the descriptions of microbe entities were mostly canonical. Dictionary matching was sufficient to solve the problem. However, for disease entities, authors prefer to use various abbreviations, and diseases are preceded by a wide variety of modifiers. Moreover, some diseases are not uniquely described. Therefore, we designed a model based on BiLSTM-CRF [28] to recognize disease entities.

For a given sentence, the concatenation of word embedding, character-based representation, and dictionary-matching representation was used as a representation for each token. For word embedding, BERT [27] is one of the most widely used and effective methods. However, the original model BERT BASE is not well applied to disease entity recognition. Therefore, we fine-tuned BERT on NCBI-disease [26] and BC5CDR-disease [29] to make its generated word embedding better at capturing semantic information in our task. In addition, we generated a character-based representation for each token using a CNN model to obtain the information of character-level regularities, such as prefixes. Finally, dictionary-matching representation aimed to obtain an indication of position of each token

in a matched word sequence. The spliced results were fed into BiLSTM-CRF for training. The effect of entity annotation is shown in the example column of Table 3.

**Table 3.** Example of entity relationship recognition.

| Example | Method |
| --- | --- |
| <m>Klebsiella pneumoniae</m> is a common cause of antimicrobial-resistant <d>opportunistic infections</d> in hospitalized patients. | Cartesian product |
| <m>Shigella</m> is a highly prevalent bacterium causing acute <d>diarrhea</d> and <d>dysentery</d> in developing countries. | Cartesian product |
| Recent studies have suggested that <m>escherichia coli</m> and <m>klebsiella pneumoniae</m>, which both cause common <d>extraintestinal infections</d> such as <d>urinary tract and bloodstream infections</d>, may also be foodborne. | Cartesian product |
| For example, <d>typhoid fever</d> is caused by the <m>capsulated salmonella enterica serovar typhi</m>, while <m>nontyphoidal salmonella serovars</m> associated with <d>gastroenteritis</d> are non-capsulated. | Clustering syntactic analysis |

In the entity relationship identification section, we present a Bi-LSTM network with entity-aware attention using latent entity typing [30] to train the relationship classification model.

Self-attention has been widely applied to many NLP tasks, such as machine translation, language understanding, and semantic role labeling. In this work, we adopted the multi-head attention formulation, which is one of the methods for implementing the self-attention. Given a matrix of n vectors, query Q, key K, and value V, the scaled dot-product attention was calculated using the following equation:

$$\text{Atention}(Q, K, V) = \text{softmax}\left(\frac{QK^\top}{\sqrt{d_w}}\right)V \tag{1}$$

In Equation (1), $d_w$ is the dimension of the vector. In this work, query Q, key K, and value V were equivalent to the word representation vectors X. As shown in Figure 5, after passing through Bi-LSTM, the final sentence representation z, resulting from this attention mechanism, was computed as follows:

$$u_i = \tanh\left(W^H\left[h_i; p_i^{e_1}; p_i^{e_2}\right] + W^E[h_{e_1}; t_1; h_{e_2}; t_2]\right) \tag{2}$$

$$\alpha_i = \frac{\exp\left(v^\top u_i\right)}{\sum_{j=1}^{n} \exp\left(v^\top u_j\right)} \tag{3}$$

$$z = \sum_{j=1}^{n} \alpha_i h_i \tag{4}$$

In the above equations, $W^H\left[h_i; p_i^{e_1}; p_i^{e_2}\right]$ represents relative position features and $W^E[h_{e_1}; t_1; h_{e_2}; t_2]$ represents entity features with latent types.

Figure 6 illustrates the results of self-attention in the sentence, 'the <e1> disease </e1> was caused by the <e2> bacterium </e2>'. There were visualizations of the two heads in the multi-head attention applied for self-attention. The color density indicated attention values, the results of Equation (1), which implies how much an entity focuses on each word in a sentence.

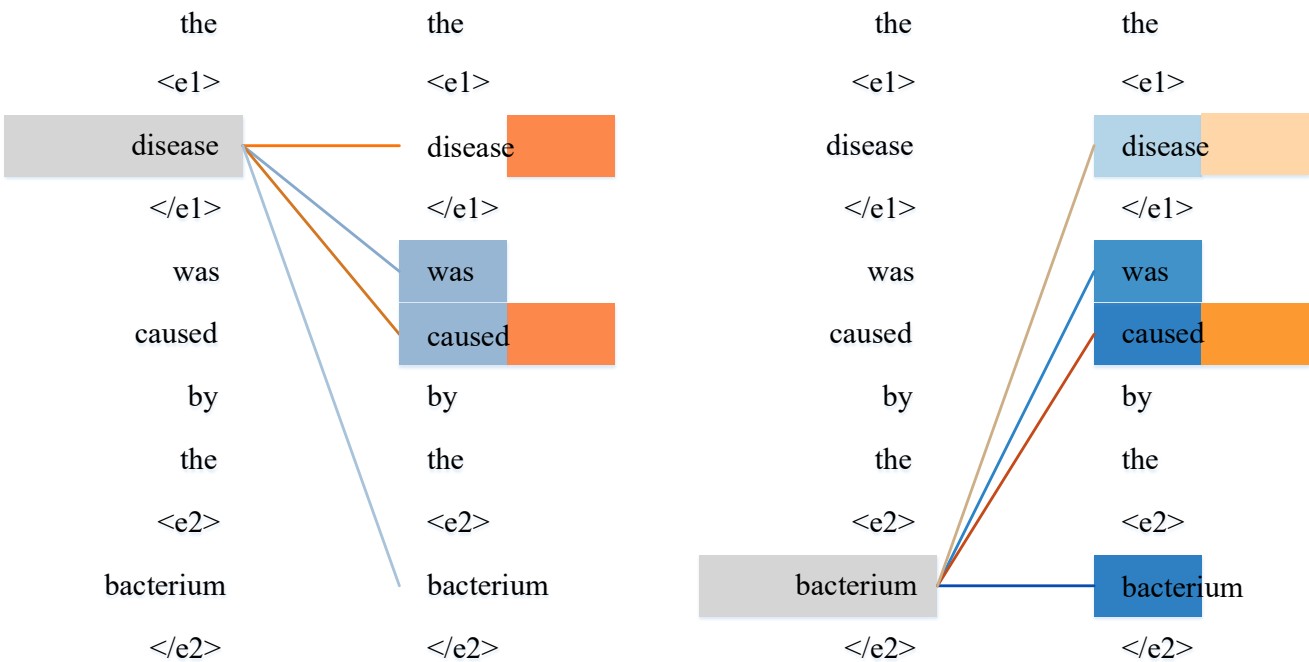

**Figure 6.** Visualization of self-attention.

In order to ensure the accuracy of GDReBase in practical applications, we designed strict artificial rules to further filter the obtained results. The core criterion was that there needs to be at least one trigger word between microbe entity and disease entity.

Of the 13,553 pieces of evidence we extracted, 90.7% belonged to one-to-one or one-to-many relationships, and such cases were called simple sentences. The first two examples in Table 3 are both simple sentences. For simple sentences, the set of diseases and the set of microbes make Cartesian products whenever a suitably placed trigger word exists. The remaining 9.3% of the evidence were all many-to-many cases, called complex sentences. When dealing with complex sentences, entities of the same category that occur consecutively are treated as a whole. Of the total, 9% were the same as the third sentence in Table 3, all of which had an even number of wholes. The Cartesian product was applied after slicing the sentences. Additionally, 0.3% of the cases were similar to the fourth example, all of which had an odd number of wholes. Syntactic analysis and clustering based on the position of the entity in the sentence are required.

*4.5. NER Evaluation Metrics*

NER systems are usually evaluated by comparing their outputs against human annotations. The comparison can be quantified as either an exact match or relaxed match. The method used in this article was an exact match.

NER involves identifying both entity boundaries and entity types. With 'exact-match evaluation', a named entity is considered correctly recognized only if both its boundaries and type match are ground truth. Precision, Recall, and F1-score are computed on the number of true positives (TP), false positives (FP), and false negatives (FN).

1. TP: entities that are recognized by NER and match the ground truth;
2. FP: entities that are recognized by NER but do not match the ground truth;
3. FN: entities annotated in the ground truth that are not recognized by NER.

Precision measures the ability of a NER system to present only correct entities, and Recall measures the ability of a NER system to recognize all entities in a corpus. F1-score is the harmonic mean of precision and recall, and the balanced F1-score is most commonly used:

$$\text{Precision} = \frac{TP}{TP + FP} \tag{5}$$

$$\text{Recall} = \frac{TP}{TP + FN} \tag{6}$$

$$\text{F1-score} = 2 \times \frac{\text{Precision} \times \text{Recall}}{\text{Precision} + \text{Recall}} \tag{7}$$

**Author Contributions:** Conceptualization, Y.X.; methodology, Y.X.; software, H.X., X.D., C.L. and X.L.; validation, H.X., X.L., D.W. and Z.S.; formal analysis, H.X. and X.L.; investigation, S.L., Y.Z., C.Z. and S.C.; resources, K.L., C.Z. and X.L.; data curation, K.L., C.Z. and X.L.; writing—original draft preparation, H.X. and K.L.; writing—review and editing, H.X.; visualization, H.X.; supervision, Y.X.; project administration, Y.X. All authors have read and agreed to the published version of the manuscript.

**Funding:** This research received no external funding.

**Institutional Review Board Statement:** Not applicable.

**Informed Consent Statement:** Not applicable.

**Data Availability Statement:** Data and evaluation scripts can be provided by the authors upon reasonable request.

**Conflicts of Interest:** The authors declare no conflict of interest.

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
