# Peer review of "GDReBase: A Knowledge Base for Relations between Human Gut Microbes and Diseases Based on Deep Learning"

_applsci, doi:10.3390/app13031614_

Round 1

Reviewer 1 Report

The authors present a text mining approach to identify papers dealing with diseases that are related bacteria species from the gut. The approach is very interesting, however, some things should be improved in the text as well as the database before publication:

- It is confusing that only organisms from the gut are included in the database especially, since the studies of the database are not clearly separated from microbiome studies. However, in the introduction are also microbiome studies cited. If GDReBase is also a ressource for microbiome studies such studies should be marked and it should be possible to search for such studies. If microbiome studies of the gut are not of special interest, the integration of bacteria from other tissues would inrease the applicability of the database a lot. At least it should be made clear what the authors are aiming at and what is the main application of the database.

- Introduction line 35-37ff: "Moreover, gut microbial dysfunction can lead to a number of neuro-degenerative diseases, such as Alzheimer’s disease, Parkinson’s disease and amyotrophic lateral sclerosis."
It is not yet clear if bacteria can really cause these disseases of if this is a coincidence. This should be modified.

- Is is not clear what exactly is the evidence data set. Is it a bechmark? Please include a clear description how this data set was generated. It is also not clear how the quality of the identification of the entitiy relation was measured. How was the F1 score calculated? Please describe clearly the procedure in the methods section.

- The comparison with other databases is described in the discussion. It would be better to have the relation to other databases in the introduction to have a better understanding of the novelty and focus of GDReBase (see above).

-  Figure 6. : The figure needs more explanations to be understood. What is the meaning of the colors?

- The methods could be explained in more detail: which journals were selected? Why bioinformatics journals? How was the quality of recognition (F1 score) determined (see above).

- Web database: If you include the character ' in the relation search,  an "internal server error" appears. The processing of special characters with SQL relevance should be checked carefully.

Author Response

Thank you for your report. Please see the attachment.

Reviewer 2 Report

From my point of view, the article is in general well written. The authors propose a knowledge base of consistently curated relationships between human gut mi-15 crobes and disease, which can continuously and automatically collecting papers published in mainstream journals by using deep learning and NLP methods for entity relationship identification.

However, there are still some issues that need to be addressed:

1. How the novelty is reflected in the method or model used for GDReBase construction? Maybe some general description of the method novelty shall be added in Section 4?

2. I suggest more details of the Fig. 5 should be described, for example, the considerations of the model architecture design, the equations used in Fig 5.

3. In line 217, Section “5. Patents” has an empty body? What does this mean?

4. Now, user can only view the database content by search some keywords (microbe or disease). The function of data browse or data list page is suggested to be provided.

I consider that it is an article with adequate quality to be published after the proposed corrections.

Author Response

(The authors gave the same response as above.)

Round 2

Reviewer 1 Report

There are now new mistakes and inaccuracies in the changed text of the manuscript (and the answer to the referees). The complete text should be checked by a native speaker.

e.g.: "Our goal is to obtain a constant flow of new and possible relationships between gut
microbes and human disease from mainstream journals."

-> Mainstream journals Provide a broad overview of topics and are written for non-scientist. Hopefully the data will be extracted from scientific journals.

"Users can use GDReBase to view the latest research findings and explore new possibilities."

-> Explore new possibilities? Please explain.

Author Response

The authors would like to thank the reviewer for his revision of the paper, for his comments and his suggestions for improvement of the manuscript. The reviewer's comments are answered in the attached file.

Reviewer 2 Report

The author has answered my questions. I think the manuscript can be accepted as its current form.

Author Response

Thank you for your kind work and review report!